# Titin’s Intrinsically Disordered PEVK Domain Modulates Actin Polymerization

**DOI:** 10.3390/ijms26147004

**Published:** 2025-07-21

**Authors:** Áron Gellért Altorjay, Hedvig Tordai, Ádám Zolcsák, Nikoletta Kósa, Tamás Hegedűs, Miklós Kellermayer

**Affiliations:** 1Department of Biophysics and Radiaton Biology, Semmelweis University, 1085 Budapest, Hungary; aron.altorjay@gmail.com (Á.G.A.); tordai.hedvig@semmelweis.hu (H.T.); zolcsak.adam@semmelweis.hu (Á.Z.); kosa.nikolett@semmelweis.hu (N.K.); tamas@hegelab.org (T.H.); 2HUN-REN-SU Biophysical Virology Research Group, Semmelweis University, 1085 Budapest, Hungary

**Keywords:** titin, PEVK, pyrene actin, polymerization assay, atomic force microscopy, AlphaFold, intrinsically unstructured protein domain, supported lipid bilayer, actin paracrystal

## Abstract

The multi-domain muscle protein titin provides elasticity and mechanosensing functions to the sarcomere. Titin’s PEVK domain is intrinsically disordered due to the presence of a large number of prolines and highly charged residues. Although PEVK does not have canonical actin-binding motifs, it has been shown to bind F-actin. Here, we explored whether the PEVK domain may also affect actin assembly. We cloned the middle, 733-residue-long segment (called PEVKII) of the full-length PEVK domain, expressed in *E. coli* and purified by using His- and Avi-tags engineered to the N- and C-termini, respectively. Actin assembly was monitored by the pyrene assay in the presence of varying PEVKII concentrations. The structural features of PEVKII-associated F-actin were studied with atomic force microscopy. The added PEVKII enhanced the initial and log-phase rates of actin assembly and the peak F-actin quantity in a concentration-dependent way. However, the critical concentration of actin polymerization was unaltered. Thus, PEVK accelerates actin polymerization by facilitating its nucleation. This effect was highlighted in the AFM images of F-actin–PEVKII adsorbed to the supported lipid bilayer. The sample was dominated by radially symmetric complexes of short actin filaments. PEVK’s actin polymerization-modulating effect may, in principle, have a function in regulating sarcomeric actin length and turnover. Altogether, titin’s PEVK domain is not only a non-canonical actin-binding protein that regulates sarcomeric shortening, but one that may modulate actin polymerization as well.

## 1. Introduction

Titin [1], also known as connectin [2], is a giant filamentous protein found primarily in vertebrate striated muscles [3]. The titin molecule spans from the Z-line, where its N-terminus is located, along the surface of the thick filament [4,5], to the M-band of the sarcomere, where its C-terminus is anchored [6]. Titin is by far the major source of passive elasticity in the striated muscle sarcomere [7], and it also plays a significant role in mechanosensing [8] and provides a template for molecular interactions [9]. Titin comprises predominantly ~300 globular domains categorized into immunoglobulin (Ig) and fibronectin (FN) classes and contains a kinase domain near the M-band, and interspersed unique sequences are also found in the molecule [10]. The largest of the unique sequences is the proline-, glutamate-, valine-, and lysine-rich PEVK domain which, in the longest skeletal isoform, contains 2174 amino acids [11]. Titin has been shown in numerous previous publications to bind actin filaments [12,13,14,15,16,17,18,19]. There are distinct regions along titin that have been shown to display actin-binding propensities: (1) a region near the Z-line where it is tightly associated with a ~100 nm long stretch of the thin filament [20,21], (2) the N2A unique sequence [22,23], (3) the PEVK domain [24,25,26,27], and (4) a series of domains in the D-zone super-repeat region [28,29]. The binding of PEVK to actin is of particular interest, as it emerged to have a potential role in regulating sarcomeric contraction [17,26,30,31,32,33].

The PEVK domain, due to its propensity of charged (E, K) and proline (P) residues, is thought to possess intrinsically unstructured properties [10]. It behaves as an entropic spring [20] with a tunable persistence length [32] averaging around 1 nm [34]. The PEVK domain has been shown to possess calcium-binding properties [35], a feature that may also have functional and regulatory implications. PEVK displays a repetitive pattern of structural motifs rich in either glutamates or the PPAK sequence element (polyE and PPAK motifs) [36,37]. The polyE and PPAK motifs have also been shown to directly bind actin, and their interaction requires a few tens of picoNewtons to break [38]. Given that PEVK conformationally responds to changes in ionic strength [32], calcium concentration [35], and pH [39], and it binds to F-actin, its role is likely to extend much beyond providing passive elasticity to the sarcomere. Although the PEVK domain lacks canonical structural elements of actin-binding proteins (ABPs) [40,41,42], the fact that many ABPs also influence actin polymerization raises the intriguing possibility that PEVK may similarly interfere with actin assembly. Therefore, in the present work, we explored the effect of a recombinant PEVK fragment on actin polymerization followed by the pyrene assay. We find that PEVK enhances actin assembly by facilitating the seeding process, but the steady-state polymerization kinetics remain unaltered.

## 2. Results and Discussion

### 2.1. Titin PEVK Accelerates Actin Polymerization

In the present work, we investigated the effect of titin’s intrinsically disordered PEVK domain on the polymerization of actin. The PEVK molecule used was a recombinant fragment of the full-length human titin corresponding to the middle one-third of the domain containing 733 amino acids [26,32] (Figure 1a). Notably, this PEVK segment is rich in polyE motifs and altogether contains 191 (26%) glutamate residues. With the help of His- and Avi-tags engineered to the N- and C-termini of the fragment, respectively, we could prepare the protein to high purity (Figure 1b).

To explore the effect of PEVKII on actin assembly, we first polymerized pyrene-labeled actin in the presence of PEVKII for 800 s and measured the fluorescence excitation and emission spectra (Figure 2a,b). When polymerized in the presence of PEVKII, both spectra displayed 16% greater intensities than in the control, indicating that the addition of PEVKII to actin resulted in a shift towards F-actin. Thus, PEVKII is able to enhance the polymerization of actin. To further investigate the effect of PEVKII on actin assembly, we followed pyrene-actin fluorescence as a function of time (Figure 2c). We found that the fluorescence intensity in the presence of PEVKII exceeded that in the control sample across the entire 800 s time period investigated, but to varying degrees depending on the timepoint: the largest difference between the traces was observed between 200 and 400 s following the launch of actin assembly.

Next, we studied the effect of changing PEVKII concentration on the polymerization of actin (Figure 3). Across a time span of 2000 s, we investigated the effects of PEVKII added at final concentrations ranging between 3.25 and 32.5 nM. We observed that PEVKII enhanced the ratio of F-actin in the initial 1500 s of the polymerization process, but following the establishment of steady-state conditions, the fluorescence intensities became comparable. The steady decrease in fluorescence intensity, seen in the PEVKII-actin samples after 500 s, is attributable to photobleaching caused by the continuous illumination of the sample. To quantitate the PEVKII-concentration-dependent effects, based on the fluorescence intensities at different time points (Figure 3a–c), we calculated the initial (Figure 3d) and half-maximal (Figure 3e) actin assembly rates, the peak intensity that reflects the maximal F-actin quantity reached (Figure 3f), and the time necessary to reach this peak (Figure 3g). The initial actin polymerization rate was increased more than 30-fold upon the addition of PEVKII in an essentially step-wise function (Figure 3d).

The result suggests that PEVKII significantly accelerates the initial, nucleation phase of actin assembly already at a concentration that is three orders of magnitude lower than that of actin. The addition of PEVKII resulted in enhancing the half-maximal actin assembly rate, too (Figure 3e), which corresponds to the rate of the log phase of the polymerization. The enhancement, however, was only six-fold, reached by raising the PEVKII concentration to 32.5 nM. The peak fluorescence intensity, which reflects the total amount of F-actin, was increased only by 22% upon PEVKII addition (Figure 3f), pointing at the possibility that upon reaching steady-state conditions, the globular-to-filamentous actin ratios are similar with or without PEVKII. The peak fluorescence intensity was reached more than three times faster when PEVKII was added (Figure 3g). Altogether, the results indicate that PEVKII accelerates the initial phases of actin polymerization, but the steady-state kinetics remain unaltered.

To dissect the effect of PEVK on the various phases of actin assembly, we varied the time point of PEVKII addition during the polymerization process (Figure 4). When PEVKII was added at the start, rapid assembly was observed. When added 500 s following the start, during the log phase, the assembly rate suddenly increased 3.4-fold. By contrast, when added 1500 s after the start, when approaching steady state, the assembly rate became slightly reduced (0.67-fold). Altogether these results support the notion that PEVK influences only the initial phases of actin assembly.

### 2.2. PEVK Facilitates Actin Nucleation While Leaving Steady-State Assembly Rates Unaffected

To investigate the effect of PEVK on steady-state actin polymerization, we carried our actin-concentration-dependent measurements (Figure 5). Pyrene-actin fluorescence emission intensity was increased in all phases of the assembly upon increasing actin concentration but keeping the PEVKII level constant (Figure 5a). To alleviate the problem of photobleaching during the extensive time-dependent measurements necessitated by the use of low actin concentrations, we carried out intermittent spectroscopic measurements (one measurement every 10 min) (Figure 5b). The steady-state fluorescence intensity versus actin concentration relation (Figure 5c) was essentially identical with and without PEVKII, indicating that the steady-state actin assembly kinetics indeed remain unaltered.

To uncover the structural features of the PEVKII–F-actin complex, we carried out AFM measurements (Figure 6). In the control F-actin sample, we observed large areas of filaments oriented in parallel, forming paracrystals (Figure 6a). The length of the actin filaments in these nematic regimes exceeded 2 µm. The two-dimensional fast Fourier transform (FFT) (Figure 6b) of the AFM image revealed periodicities that correspond to the inter-filament distance (13.3 nm) and its harmonics. By contrast, in the AFM image of the PEVKII–F-actin sample, we found shorter filaments (<1 µm) that were arranged in radially symmetric regions as if the filaments emerged from a common center of origin (Figure 6c). The 2D FFT of the AFM image revealed no distinct periodicities. Nematic regions could not be found at smaller magnifications (greater fields of view) either (Figure 6e,g), and the 2D FFT functions were devoid of local spatial frequency maxima (Figure 6f,h), indicating the lack of spatial periodicities.

The PEVKII–F-actin samples were scattered with globular structures, which were often located in the focal centers of the filaments. Conceivably, these globular structures correspond to the PEVKII molecules (or multimers thereof) forming seeds of actin polymerization. The AFM results thus support the findings that PEVK enhances only the early, nucleation phase of actin assembly, by providing seeds from which numerous filaments may emerge. When adding PEVKII during the log phase of actin assembly (see Figure 4), new seeds may form from the G-actin pool still present. However, upon approaching the steady state, where the G-actin pool has been largely exhausted, the addition of further PEVKII is no longer able to generate seeds of polymerization, leaving the steady-state kinetics unaltered.

### 2.3. PEVK Domain as a Local Actin Modulator

Our results show that the PEVK domain of titin accelerates the polymerization of actin primarily by facilitating the seeding of its assembly. It is an intriguing question how the PEVK domain may elicit this function in situ in the sarcomere. We need to address the following specific questions: (1) What is the structure of PEVK in the actin-associated and dissociated states? (2) How accessible is actin to the PEVK binding? (3) What is the role of PEVK binding in the in situ actin assembly dynamics?

First, since the identification of titin’s sequence and the discovery of its unique sequence element, the PEVK domain has been thought to acquire an intrinsically disordered, random structure that displays entropic elastic properties [10,20]. Indeed, AlphaFold predicted a completely disordered structure for this domain supported by low pLDDT confidence scores (Figure 7a), and previous observations suggest that PEVK behaves as an entropic chain with a persistence length (*L_P_*) of ~1 nm [32,34]. Based on this persistence length and the 826 nm contour length (*L_C_*) of the domain (calculated from the 2174 amino acids [10] and 0.38 nm residue spacing [43]), the mean end-to-end distance (*R*) of the conformationally relaxed PEVK, calculated as [44](1)$$R=\sqrt{L_{C}L_{P}}$$
is 28.7 nm. Based on the obtained end-to-end distance, the radius of gyration (*R_G_*), which is the radius of the theoretical sphere that contains the entire mass of the PEVK domain, obtained as [45](2)$$R_{G}=\frac{R}{\sqrt{6}}$$
is 11.7 nm. The PEVK domain, as an intrinsically disordered protein, lacks the structural features of canonical actin-binding proteins [40,41,42]. However, it is possible that upon binding to the thin filament, PEVK acquires a more stable structure, which is a characteristic feature of intrinsically unstructured domains [46]. It is quite plausible that the large experiment-to-experiment and sample-to-sample variation in our results (cf. Figure 2c, Figure 3a, and Figure 5b) is due to a variation in the actual structure attained by PEVK upon binding to actin. The details of this induced structure are to be investigated in further experiments. While the exact molecular mechanisms of PEVK binding to actin are not known, we hypothesize that the polyproline-rich regions of PEVK [24,47] may associate with actin similarly to the way formin’s FH1 domain binds to profilin–actin via its polyproline tracks [48,49,50,51].

Second, the accessibility of PEVK to actin binding is likely to be influenced by the contractile status and the lattice spacing of the sarcomere (Figure 7b,c). The conformation, and hence the effective space occupied by PEVK, is affected by the thick/thin filament overlap [26]. Upon increasing the overlap, PEVK becomes more contracted. It reaches its most contracted state, and hence the highest entropy state, as the sarcomere approaches its slack length (Figure 7b). Notably, the effective concentration of the actin-binding PEVK motifs is the largest in this arrangement, and as a result the rate of binding to actin reaches the maximum. In this relaxed configuration, the overall PEVK domain likely resembles a sphere with a radius of 11.7 nm (corresponding to the radius of gyration, *R*_G_) (Figure 7c).

Accordingly, considering the filament spacing in the I-band [52], the thin filaments are surrounded by a cloud of the PEVK domain (Figure 7c). Altogether, there is ample opportunity for the PEVK domain to bind to actin under in situ conditions. Furthermore, via dynamically modulating the conformation, and hence the space occupied by PEVK, the domain’s binding to actin is regulated by the contractile status of the sarcomere. The question that prevails is what is the mechanism by which PEVK facilitates actin nucleation? It is not inconceivable that the poly-proline sequences of the PEVK domain play a role in the process, similar to how formin nucleates the growth of the actin–profilin complex [48,49,50,51].

Finally, an intriguing question is whether the binding of PEVK to actin in situ results in an effective modulation of its assembly dynamics. It is known that actin is dynamically turned over in the sarcomere on a time scale of a day [53,54,55]. Numerous actin-binding proteins play a role in the processes of actin turnover and thin-filament length regulation, although the molecular mechanisms are far from being understood. It appears, however, that factors mediating the assembly of filaments from monomers and catalyzing their elongation are important in the regulation [56]. It is conceivable that in the crowded environment of the sarcomeric lattice, the PEVK domain, with its dynamically varying and adaptable conformation, and by its actin-binding and nucleation capacities, contributes significantly to the process.

## 3. Materials and Methods

### 3.1. PEVKII Cloning, Expression and Purification

The middle one-third segment of the human skeletal muscle PEVK domain (named PEVKII [26]), encompassing nucleotide sequences 33043–35325 (GenBank Sequence ID number NM_001267550.2), was cloned into a pet28a vector. To facilitate purification, His- and Avi-tags were appended to the N- and C-termini of the recombinant protein, respectively. PEVKII was expressed in *E. coli* BL21 (DE3)pLysS cells. After lysing the cells, protein purification was achieved from the soluble fraction by the sequential use of Ni^2+^-NTA (for capturing via the N-terminus) and Pierce™ Monomeric Avidin Agarose (for capturing via the C-terminus) columns under native conditions following manufacturer’s protocols. The tags were not cleaved from PEVKII. However, based on our prior publications [26,32,38] and literature data [57], it was assumed that neither the His- nor the Avi-tag binds actin or interferes with its polymerization. To estimate the concentration of PEVKII, SDS-PAGE was performed on a 10% gel with 5 µL of PEVKII and increasing amounts of bovine serum albumin (BSA, 50–1000 ng), used for calibration, loaded on the gel (see Figure 1). Subsequently, the gel was stained with Coomassie brilliant blue, and densitometric analysis was performed by using ImageJ (v. 1.53, public domain, Wayne Rasband, NIH). PEVKII concentration in the given sample was then obtained from the calculated total protein mass divided by the loaded-sample volume.

### 3.2. Actin Purification and Pyrene Labeling

Actin was purified from acetone-extracted muscle powder prepared from rabbit *m. longissimus dorsi* according to procedures published earlier [58,59,60,61]. Actin extraction and purification from the muscle powder involved two cycles of polymerization and depolymerization. To follow the polymerization of actin, we used pyrene-labeled actin [62,63]. To carry out the pyrene labeling, N-(1-pyrenyl)iodoacetamide was added to F-actin, polymerized in 100 mM KCl, 2 mM MgCl_2_, and 2 mM Tris-HCL (pH 8.0), at a pyrene/actin molar ratio of 1.2. Labeling was carried out overnight at room temperature in the dark. Subsequently, F-actin was sedimented (400,000× *g*, 45 min, 4 °C), then depolymerized by homogenization and dialysis in G-buffer (4 mM Tris-HCl, 0.2 mM ATP, 0.2 mM CaCl_2_, 1 mM DTT, 0.005% NaN_3_, pH 8.0), and finally clarified by centrifugation (400,000× *g*, 45 min, 4 °C). The pyrene-labeled G-actin was stored on ice and used within one day.

### 3.3. Actin Polymerization Assay

The polymerization of actin was followed by measuring the steady-state fluorescence intensity of pyrene-labeled actin as a function of time, as established before [62,63]. The ratio of pyrene-labeled and non-labeled actin was typically 10% in our experiments. Total actin concentration was usually 4 µM, except in actin-concentration-dependent measurements, where the actual concentrations are indicated. Prior to initiating polymerization, the G-actin solution, containing the pyrene-labeled and non-labeled actin in the proper ratio, was supplemented with an equal volume of Ca-Mg ion exchange buffer (50 µM MgCl_2_, 0.2 mM EGTA, pH 8.0) and incubated in a glass cuvette for two minutes. Actin assembly was initiated by adding 1/19 volume of 20 × polymerization buffer (2 M KCl, 40 mM MgCl_2_, 40 mM Tris-HCl, 1 mM EGTA, 1 mM DTT, 0.01% NaN_3_, pH 8.0). PEVKII was added to the sample so that the final buffer composition and concentrations remained consistent throughout the polymerization assay: 100 mM KCl, 2 mM MgCl_2_. The final concentration of PEVK varied between 3.25 and 125 nM. The actual concentrations used are indicated in the figure legends. Actin polymerization was followed by measuring fluorescence emission intensity at 25 °C with an FLS 980 spectrofluorimeter (l_ex_ 365 nm, l_em_ 407 nm, Edinburgh Instruments Ltd., Livingston, UK).

### 3.4. Preparation of Supported Lipid Bilayer

To image actin filaments and actin–PEVK complexes, the samples were attached to a supported lipid bilayer (SLB). To prepare the SLB, first a lipid mixture comprising dipalmitoyl-phosphatidylcholine (DPPC) and 1,2-dioleoyl-sn-glycero-3-ethylphosphocholine (DPEPC) in a 1:1 molar ratio was hydrated in a buffer solution containing 10 mM Tris-HCl, 100 mM NaCl, and 3 mM CaCl_2_, pH 7.4 [22]. The final lipid concentration was adjusted to 1 mM. Lipid vesicles were generated by extruding the hydrated lipid mixture through a polycarbonate membrane using an Avanti Mini Extruder at 55 °C with a pore size of 100 nm. A quantity of 100 µL of the suspension was added to a freshly cleaved mica surface and incubated at room temperature for 30 min. Subsequently, the temperature was raised to 65 °C for 15 min to facilitate vesicle rupture, thus forming a stable, positively charged lipid bilayer on the mica substrate.

### 3.5. Atomic Force Microscopy (AFM) Measurements

The structural and topographical characteristics of the actin filaments were analyzed by using a Cypher ES Atomic Force Microscope (Asylum Research, Oxford Instruments, Santa Barbara, CA, USA) operated in non-contact mode. Silicon cantilevers (model BL-AC40TS, Olympus, Tokyo, Japan) with a nominal tip radius of 8 nm and resonance frequency of ~25 kHz were used. Imaging was carried out in a hydrated environment to maintain the native conformation of both the lipid bilayer and the actin filaments. Pyrene-labeled G-actin was polymerized in bulk under standard conditions (actin concentration 4 µM, ratio of pyrene-labeled actin 10%, PEVKII added at the appropriate concentrations) for 1000 s. Subsequently, 20 µL of the sample was added to the LBS and incubated for 15 min. The sample surface was then rinsed gently with polymerization buffer to remove unbound filaments. AFM imaging was carried out in the polymerization buffer at 25 °C. Images having a size of 512 × 512 pixels were acquired at typical line-scan rates of 1 Hz at high setpoint values (~90% of the free cantilever oscillation amplitude) to minimize mechanical damage to the sample. AFM images were processed with the built-in algorithms of the driver software.

### 3.6. Structure Prediction

The structure of PEVKII was predicted by using AlphaFold3 (3.0.1.) [64] installed 10 March, 2025 locally following the instructions at https://github.com/google-deepmind/alphafold3, and configured to run on a system with an AMD 16-core processor (Advanced Micro Devices, Inc. (AMD), Sunnyvale, CA, USA) and an NVIDIA RTX A6000 GPU (NVIDIA Corporation, Santa Clara, CA, USA) with 48 GB of VRAM. The PyMOL Molecular Graphics System (Version 2.4 Schrödinger, LLC, New York, NY, USA) was used for analyzing and visualizing the predicted structures.

### 3.7. Calculations and Statistics

Data were analyzed, compared, and plotted by using commercial software (MS Excel, IgorPro versions 6 and 8, KaleidaGraph v.5).

## 4. Conclusions

The intrinsically disordered PEVK domain of titin is a multi-functional unit that not only binds actin but is capable of enhancing its assembly by facilitating nucleation. Under in vitro conditions, this property of PEVK results in the formation of radially symmetric actin filament complexes. The actin polymerization-modulatory effect of PEVK may conceivably have a function in regulating actin length and turnover. Thus, titin’s PEVK domain is a non-canonical actin-binding protein that modulates both sarcomere shortening and actin assembly.

## Figures and Tables

**Figure 1 ijms-26-07004-f001:**
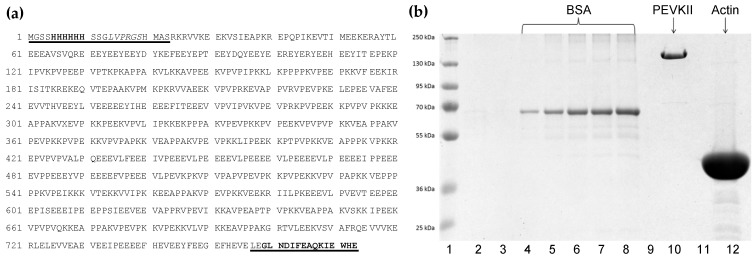
(**a**) Amino acid sequence of PEVKII. Non-PEVKII amino acids are underlined. The N-terminal His_6_-tag and the C-terminal Avi-tag sequences are shown in bold. The thrombin recognition site is indicated in italics. (**b**) SDS-PAGE of the proteins used in the experiments. Lane 1, molecular weight markers; lanes 4–8, bovine serum albumin (BSA) in increasing amounts (50, 100, 250, 500, 1000 µg), used for calibrating total protein mass; lane 10, cloned, expressed, and purified PEVKII segment. Note that even though the expected molecular weight of PEVKII is ~80 kDa, its band is positioned above 130 kDa because it is highly charged (net charge −64); hence, its electrophoretic motion is retarded.

**Figure 2 ijms-26-07004-f002:**
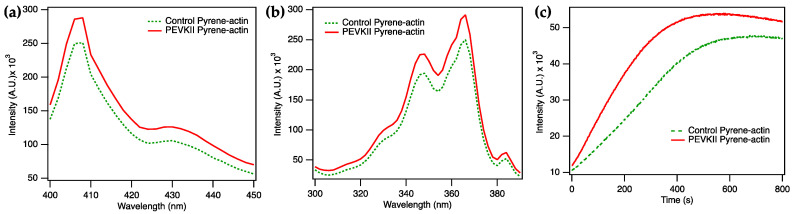
Spectroscopic analysis of pyrene-labeled actin polymerized in the presence of PEVKII. (**a**) Fluorescence emission spectra (λ_ex_ 365 nm) recorded after 800 s of actin polymerization. (**b**) Fluorescence excitation spectra (λ_em_ 407 nm) recorded after 800 s of actin polymerization. (**c**) Time-dependent changes in fluorescence emission, measured at 407 nm (λ_ex_ 365 nm) across a time span of 800 s. Actin concentration was 4 µM, and the ratio of pyrene-labeled actin was 10%. PEVKII concentration was 5.5 nM. Green dotted and red continuous lines indicate control (actin only) and PEVKII-dependent traces, respectively.

**Figure 3 ijms-26-07004-f003:**
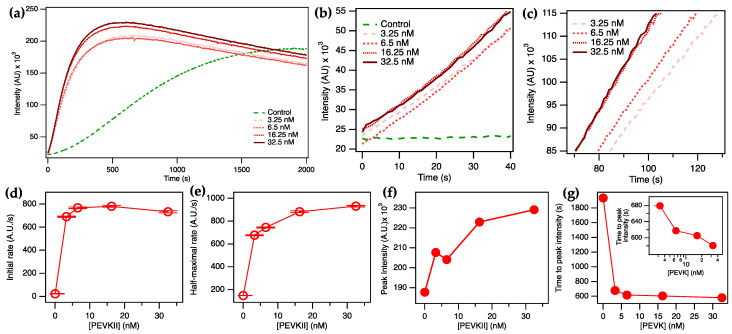
Effect of PEVKII on actin polymerization (4 µM actin, 10% pyrene-labeled). (**a**) Time trace of fluorescence emission (λ_ex_ 365 nm, λ_em_ 407 nm) across a span of 2000 s. (**b**) Zoomed-in part of the trace in the time window of 0–40 s, which reflects the initial assembly rates. (**c**) Zoomed-in part of the trace in the time window of 70–130 s, which accommodates the region in the half-maximal fluorescence emission intensity. (**d**) Initial rate of actin polymerization as a function of PEVKII concentration. Error bars correspond to the error of linear fit. (**e**) Half-maximal actin polymerization rate as a function of PEVKII concentration. Error bars correspond to the error of linear fit. (**f**) Peak F-actin quantity as a function of PEVKII concentration. (**g**) Time to peak fluorescence emission intensity, as a function of PEVKII concentration. Inset shows the same function on the logarithmic scale.

**Figure 4 ijms-26-07004-f004:**
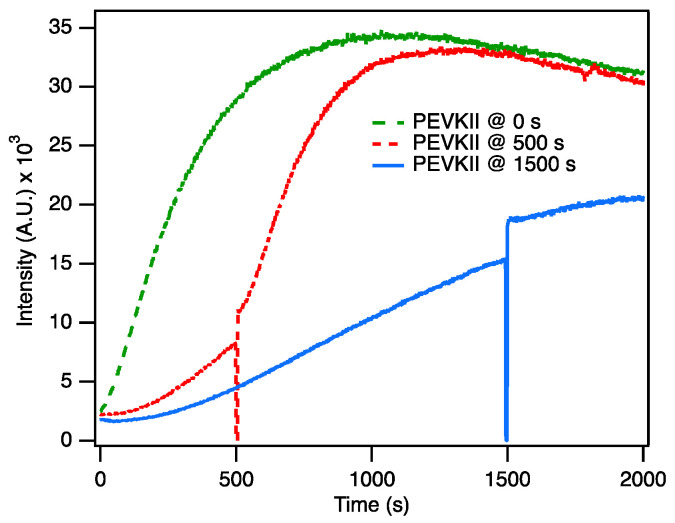
Effect of PEVKII on ongoing actin polymerization (4 µM actin, 10% pyrene-labeled). PEVKII was added to the same final concentration (125 nM), but at different times of the polymerization process: at the start (0 s, green segmented line), after 500 s (red dotted line), and after 1500 s (blue continuous line). When PEVKII was added after 500 s, the rate of actin polymerization increased suddenly 3.4-fold. By contrast, when added after 1500 s, actin polymerization rate slightly decreased (0.67-fold).

**Figure 5 ijms-26-07004-f005:**
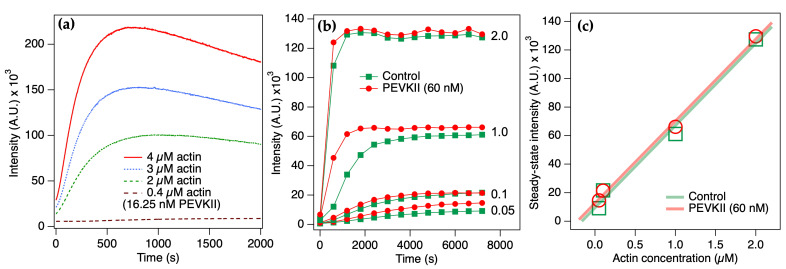
Effect of actin concentration on the PEVKII-induced changes. (**a**) Time trace of fluorescence emission (λ_ec_ 365 nm, λ_em_ 407 nm) across a span of 2000 s in the presence of 16.25 nM PEVKII and different concentrations of actin (0.4–4.0 µM, 10% pyrene labeled). Fluorescence intensity was measured under continuous illumination. (**b**) Time trace of fluorescence emission across a span of 8000 s in the presence of 60 nM PEVKII and different concentrations of actin (0.05–2.0 µM, 10% pyrene labeled). The sample was illuminated only during the intermittent (every 10 min) spectroscopic measurement. (**c**) Fluorescence emission intensity as a function of actin concentration in steady-state assembly conditions (after 8000 s of actin polymerization).

**Figure 6 ijms-26-07004-f006:**
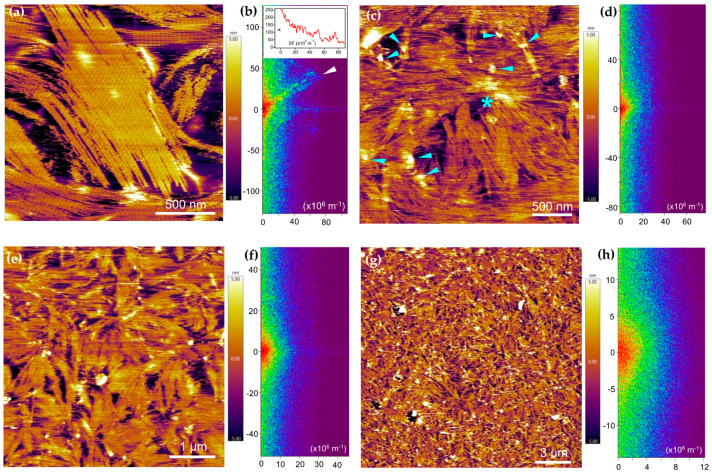
Effect of PEVK on actin filament structure and organization, measured with AFM. (**a**) Height-contrast AFM image of control actin filaments forming paracrystal on a supported lipid bilayer (SLB). A quantity of 4 µM 10% labeled pyrene actin was polymerized for 1000 s and pipetted on the SLB (DPPC:DPEPC, 1:1 molar ratio). (**b**) Two-dimensional fast Fourier transform (2D FFT) of the control F-actin AFM image. Inset, amplitude (*A*) versus spatial frequency (*SF*) plot along the direction indicated by the white arrowhead in the FFT. The peaks in the spectrum indicate periodicities ranging between 13.3 nm and 200 nm (corresponding to SF values between 75 × 10^6^–5 × 10^6^ m^−1^). (**c**,**e**,**g**) Height-contrast AFM images of actin filaments polymerized in the presence of PEVKII (11 nM). Light-blue asterisk indicates a focal point from which filaments radiate. Light-blue arrowheads point at granular particles that likely correspond to PEVKII molecules. (**d**,**f**,**h**) Two-dimensional FFT images of the corresponding F-actin–PEVKII AFM images.

**Figure 7 ijms-26-07004-f007:**
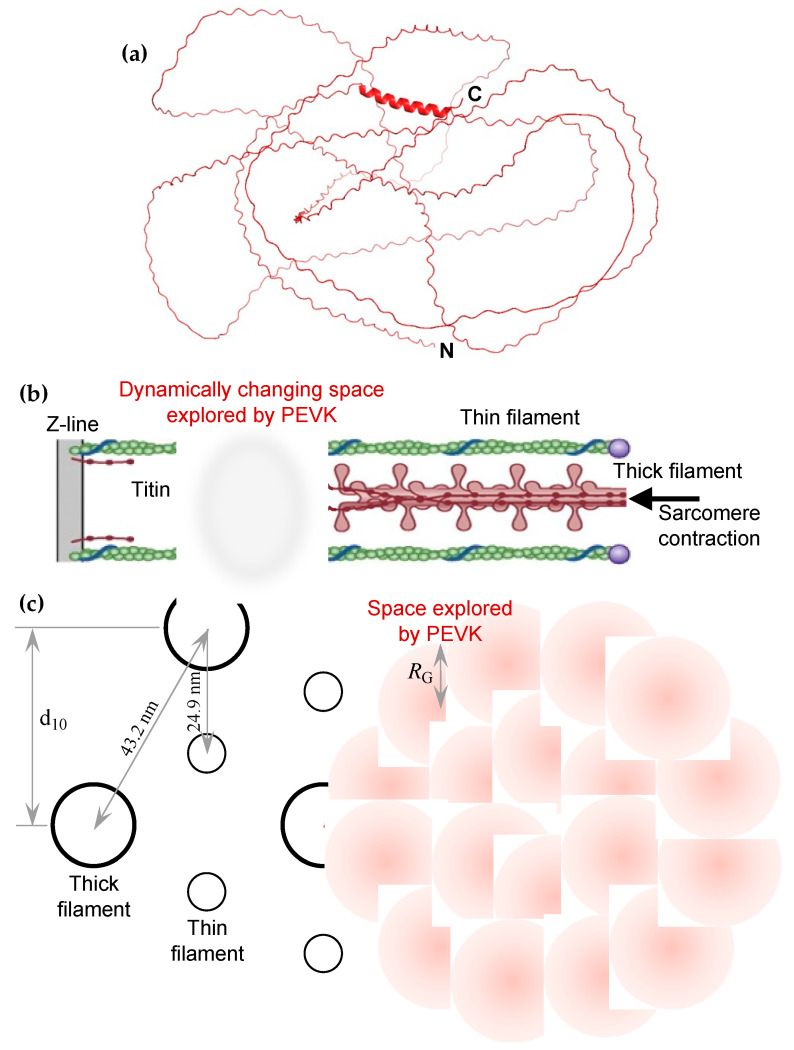
(**a**) Structure of PEVKII predicted by AlphaFold3. An essentially completely disordered structure is predicted, revealed by low confidence scores (98% of residues possess pLDDT scores lower than 70). The short alpha helical region corresponds to the stretch of the peptide between Asp73 and Glu90. (**b**) Schematic model of how the intrinsically unstructured PEVK may interact with actin in the sarcomere. The oval indicates the approximate region explored by the PEVK domain. (**c**) Schematics of the cross-sectional lattice in the overlap region (**left**) and the I-band (**right**) of the sarcomere. d_10_ is the thick-filament lattice spacing (37.4 nm) in the 1.0 x-ray-crystallographic plane, and *R*_G_ is the radius of gyration (11.7 nm) of the full-length PEVK domain. The two dotted lines correspond to the axes, projected in the transverse sarcomeric plane, of titin molecules interconnecting the tip of the thick filament and the surface of the thin filament (reached at ~100 nm from the Z-line), assuming that each of the six titins running along the thick-filament surface binds to a different thin filament.

## Data Availability

All data are available upon prior request to the corresponding author.

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
