# Peer review of "Titin’s Intrinsically Disordered PEVK Domain Modulates Actin Polymerization"

_ijms, 2025, doi:10.3390/ijms26147004_

Round 1
Reviewer 1 Report
Comments and Suggestions for Authors
PEVK actin binding paper review
This paper describes a series of experiments characterizing how the PEVK region of titin modulates actin polymerization. The PEVK region has been shown to bind actin filaments in previous studies so it was hypothesized that this region might impact actin cycling as well. The authors recombinantly expressed and purified a 733 amino acid construct that consists of the sequence composing the middle of the PEVK (PEVK II). They achieved a high level of purity for their construct and used this to explore how actin polymerization might be modulated by the PEVK II region. Using a combination of AFM and fluorescence, it was shown that this sequence increased polymerization and log-phase growth rates of actin filaments, but it did not exhibit a significant effect on steady-state assembly rates. Overall, this is a well-conceived and executed study and the conclusions are well supported by their data.
Major critiques:
- The PEVKII construct contains both Avi- and His-tags. In theory, these should not interact with actin, but it would be good to include data showing that similar effects are achieved without the tags. It is not clear if there were protease sites for something like TEV incorporated into the vector so clarifying this point at the very least would strengthen the paper.
Minor Critiques:
Lines 19-22 – this is a very long sentence in the abstract and it might read more smoothly if the sentence was divided into two sentences.
Line 26 – “in principle” should have a comma before and after.
Line 39 – I think it should be provides and not providing.
Line 40 – I would recommend adding the work predominantly or something similar in front of immunoglobulin domains since there are other structures besides Ig and Fn domains.
Line 94 – I would recommend altering the colors of the graphs to things other than red and green since there are red/green color blind individuals who might struggle to distinguish the lines.
Line 231 – The sentence that starts on this line talks about polyproline-rich regions but there is not a definition given. Looking at the PEVKII sequence, there are a few regions that seem obvious but there are others that could be considered polyproline so defining how the authors are using this term would help the readers to focus on the right parts of the sequence.
Author Response
Response to Reviewer 1
We thank the reviewer for taking the the time and effort to critically analyze our manuscript and provide helpful questions and comments.
Comment: This paper describes a series of experiments characterizing how the PEVK region of titin modulates actin polymerization. The PEVK region has been shown to bind actin filaments in previous studies so it was hypothesized that this region might impact actin cycling as well. The authors recombinantly expressed and purified a 733 amino acid construct that consists of the sequence composing the middle of the PEVK (PEVK II). They achieved a high level of purity for their construct and used this to explore how actin polymerization might be modulated by the PEVK II region. Using a combination of AFM and fluorescence, it was shown that this sequence increased polymerization and log-phase growth rates of actin filaments, but it did not exhibit a significant effect on steady-state assembly rates. Overall, this is a well-conceived and executed study and the conclusions are well supported by their data.
Response: We thank the reviewer for the succinct summary and the overall positive opinion.
Major critiques:
- The PEVKII construct contains both Avi- and His-tags. In theory, these should not interact with actin, but it would be good to include data showing that similar effects are achieved without the tags. It is not clear if there were protease sites for something like TEV incorporated into the vector so clarifying this point at the very least would strengthen the paper.
Response: Although our construct does contain a thrombin cleavage site after the N-terminal His-tag, we did not cleaved this tag off. There are strong literature data (e.g., Yang et al, PNAS 119, e2122420119, 2022) that demonstrate the lack of actin-binding by the His-tag, and it is commonly understood that the Avi-tag (hence the biotin tag) lacks actin-binding features altogether. The actin-binding capacity of PEVKII relies heavily on our prior published results (see Nagy et al, J. Cell Sci. 117, 5781, 2004, Nagy et al, Biophys. J. 89, 329, 2005, Bianco et al, Biophys. J. 93, 2102, 2005), in which the actin-binding properties of the PEVK domain were extensively and thoroughly analyzed. For clarification, we included a brief notion and references in the Methods section of our manuscript on this issue.
Minor Critiques:
Lines 19-22 – this is a very long sentence in the abstract and it might read more smoothly if the sentence was divided into two sentences.
Response: Thank you for pointing at the possibility of improving the text. We duly partitioned the sentence into two.
Line 26 – “in principle” should have a comma before and after.
Response: Thank you for the suggestion. In response, we made the correction.
Line 39 – I think it should be provides and not providing.
Response: Thank you for pointing out the grammatical error. In response, we made the correction.
Line 40 – I would recommend adding the work predominantly or something similar in front of immunoglobulin domains since there are other structures besides Ig and Fn domains.
Response: Thank you for the suggestion. In response, we amended the sentence.
Line 94 – I would recommend altering the colors of the graphs to things other than red and green since there are red/green color blind individuals who might struggle to distinguish the lines.
Response: Thank you for the suggestion. In response, we modified the graphs so that broken lines are also used. In this way the datasets can be distinguished based both on color and line style.
Line 231 – The sentence that starts on this line talks about polyproline-rich regions but there is not a definition given. Looking at the PEVKII sequence, there are a few regions that seem obvious but there are others that could be considered polyproline so defining how the authors are using this term would help the readers to focus on the right parts of the sequence.
Response: Thank you for the remark. The presence of polyproline sequences in titin have been documented in previous papers. Therefore, we added specific references in the sentence to clarify this matter.
Reviewer 2 Report
Comments and Suggestions for Authors
Investigation of the role of the giant protein titin in muscle contraction is an important task. The authors showed for the first time that the titin PEVK domain can modulate actin polymerization. This is certainly an interesting fact. The main question concerns how the PEVK domain participates in actin polymerization in the muscle sarcomere. The authors offer a very plausible explanation for this possibility in situ.
There are a few minor comments.
- Figure 1. (a) Amino acid sequence of PEVKII. - It would be better to indicate at what amino acid number the sequence of the fragment used begins and ends.
- Figure 2. - It is necessary to indicate what PEVKII concentration was used.
- The 3.1. PEVKII cloning, expression and purification Section - It is necessary to indicate whether the His- and Avi-tags were removed before the experiments.
- The 3.3. Actin polymerization assay Section - It is necessary to indicate what PEVKII concentration was used in the polymerization assay.
Author Response
Response to Reviewer 2
We thank the reviewer for taking the the time and effort to critically analyze our manuscript and provide helpful questions and comments.
Comment: Investigation of the role of the giant protein titin in muscle contraction is an important task. The authors showed for the first time that the titin PEVK domain can modulate actin polymerization. This is certainly an interesting fact. The main question concerns how the PEVK domain participates in actin polymerization in the muscle sarcomere. The authors offer a very plausible explanation for this possibility in situ.
Response: We thank the reviewer for the succinct summary and the overall positive opinion.
There are a few minor comments
- Figure 1. (a) Amino acid sequence of PEVKII. - It would be better to indicate at what amino acid number the sequence of the fragment used begins and ends.
Response: Thank you for the suggestion. We amended Figure 1.a with respective amino acid numbers and additional information. - Figure 2. - It is necessary to indicate what PEVKII concentration was used.
Response: Thank you for pointing out the missing information. The PEVKII concentration in this experiment was 5.5 nM, which is now indicated in the figure legend of the revised manuscript. - The 3.1. PEVKII cloning, expression and purification Section - It is necessary to indicate whether the His- and Avi-tags were removed before the experiments.
Response: The tags were not removed, and we now indicate this fact in the revised manuscript. Furthermore, we also added a brief explanation and references on why we did not hink it was necessary to cleave off the tags. - The 3.3. Actin polymerization assay Section - It is necessary to indicate what PEVKII concentration was used in the polymerization assay.
Response: The PEVK concentration varied in our experiments due to technical reasons associated with the determination of its concentration. The concentrations used are indicated in the figure legends. In the Methods section of the revised manuscript we provide the range of concentrations used.